# Evaluation of Synergetic Development of Water and Land Resources Based on a Coupling Coordination Degree Model

Cuimei Lv [1], Wenchao Xu [1,2], Minhua Ling [2,*], Sensen Wang [1,2] and Yuguang Hu [1,2]

1   Yellow River Laboratory, Zhengzhou University, Zhengzhou 450001, China
2   School of Water Conservancy and Civil Engineering, Zhengzhou University, Zhengzhou 450001, China
*   Correspondence: lingminhua111@163.com

**Abstract:** The interaction between water and land resources (WALRs) has been further enhanced with the development of human production activities. Evaluating the synergetic development (SD) level of WALRs is conducive to discovering the weakness of comprehensive utilization of resources and promoting sustainable development. However, previous studies did not clearly elucidate the effects of the synergetic development between WALRs (SD-WALRs). For evaluation methods, the impact of various subsystem development levels on the whole system is often ignored due to its unclear definition. Therefore, in this research, the concept of the SD-WALRs was defined based on synergetic theory. By using the "Driving Force–Pressure–State–Impact–Response" (DPSIR) model, comprehensive evaluation index systems of WALR development were established. The index systems were regarded as efficacy functions of the coupling coordination degree (CCD) model, and the evaluation model of the SD-WALR level was constructed based on it. Taking Luoyang City as an example, using the latest 10 years (2010–2019) with available data as the study period, the results showed that the value of the SD degree increased from 0.609 to 0.789 during the study period, which reached the level of intermediate synergetic development. In general, the development of WALRs showed a positive trend from "high coupling but low synergetic development" to "high synergetic development".

**Keywords:** water and land resources (WALRs); synergetic development (SD); coupling coordination degree (CCD) model; Luoyang City

## 1. Introduction

Water and land resources (WALRs) are the basic components of nature and the material basis that is essential to human survival and development, which play an irreplaceable role in maintaining ecosystem and society stability [1–3]. In recent years, rapid urbanization, population growth and improvement in people's living standards have made the tension between human demand and resource supply more serious. How to realize sustainable development and the utilization of WALRs has become a key issue that needs to be discussed in depth [4,5]. However, WALRs are not independent; they influence and restrict each other [6]. On the one hand, there is a natural mutual relationship, where the hydrological cycle is an important sub-cycle of the geological cycle, which plays an important role in shaping landforms, weathering rocks and material removal. Land resources are also the main carriers of water storage and change. On the other hand, water is required for almost all land-use activities, such as cultivation, mining and industrial production [7,8]. Meanwhile, the quantity, quality and distribution of water are affected by land-use structure and vegetative cover. Anthropogenic activities based on WALRs have further strengthened their mutual relationship. It is obvious that the quantity and quality of WALRs can directly affect the development and utilization of resources, which may cause the eco-economic benefits generated by production for humans to increase or decrease. Furthermore, the quantity and quality matching degree of WALRs and reciprocity are also remarkably influential factors for eco-economic benefits generation, where the unbalanced or uncoordinated

utilization of WALRs will impede sustainable development and even cause ecological degradation [9]. Therefore, realizing a high-level SD-WALRs is the foundation of regional sustainable development and environmental protection, while evaluating the level of the SD-WALRs can regulate the cooperation effect of regional WALRs, discover weaknesses in the comprehensive utilization of resources and promote sustainable development.

The complex interactions of WALRs were investigated in a large number of studies, which can be divided into two categories. One involves taking water resources or land resources as a separate research subject to discuss the impact of changes on the other one. Bronstert et al. [10] summarized how the changes in land utilization, such as agricultural activities, vegetation changes, road construction and urbanization, may affect hydrological processes. The studies of Weber et al. [11] and Farley et al. [12] showed that runoff yield would be increased when grassland expanded or forests shrunk. Xia et al. [13] and Capra et al. [14] investigated the impacts of slope erosion and underlying surface changes on runoff production, respectively. Zhang [15] pointed out that the optimal allocation of agricultural water can ensure the production capacity of cultivated land. These studies focused on the impact of changes in either water or land resources, such as physical and chemical properties and their evolution laws, that were caused by the other resource. Although they had strong pertinence, the mutual relationship between water and land resources could not be analyzed from an overall perspective. The feedback between changes in WALR and the society, economy and ecology could not be fully interpreted either.

The other kind of studies investigated water resources and land resources as two distinct but closely connected subsystems based on systematic theory. These studies analyzed the interaction of WALRs in terms of coupling and coordinated relationships and took the effects of the whole system on the economy, society and eco-environment into account. For example, Tong and Chen [16] coupled land utilization with surface water quality to analyze their bidirectional relationship. Dong et al. [17] analyzed the coupling effects and spatiotemporal differentiation between the urbanization process and WALR exploitation. Lv [18] evaluated the comprehensive bearing capacity of two subsystems through the coupling coordination degree of WALRs. Tan [19], on the basis of Lv's research, proposed corresponding development modes according to various comprehensive bearing capacities. It can be seen that systematic theory could be used to analyze the interaction of WALRs overall and reflect its intensity by calculating the value of the coupling coordination degree. However, water or land resources are relative subsystems. For their collaboration effects, the mutual matching and orderly development of subsystem elements are crucial. Unbalanced, disorganized development of any subsystem will impede its efficient utilization and sustainable development. In previous studies, the effects analysis of the SD-WALRs was incomplete, which may cause people to not know what state a WALR system is in and how it can achieve high synergetic development. Furthermore, the effects of the subsystems' development level on the evolution trend of the overall system were neglected.

The effects of the SD-WALRs were defined based on synergetic theory in this study. With full consideration of the driving force and pressure from society and the economy, the state of WALR utilization under their mutual relationship and the measures of environmental protection taken by human beings, the comprehensive evaluation index systems of WALRs were constructed using a DPSIR model, which could assess the value of the subsystems development level. Then, the SD-WALR evaluation model could be constructed by taking the comprehensive evaluation index systems as efficacy functions of the CCD model. The overall research idea is shown in Figure 1.

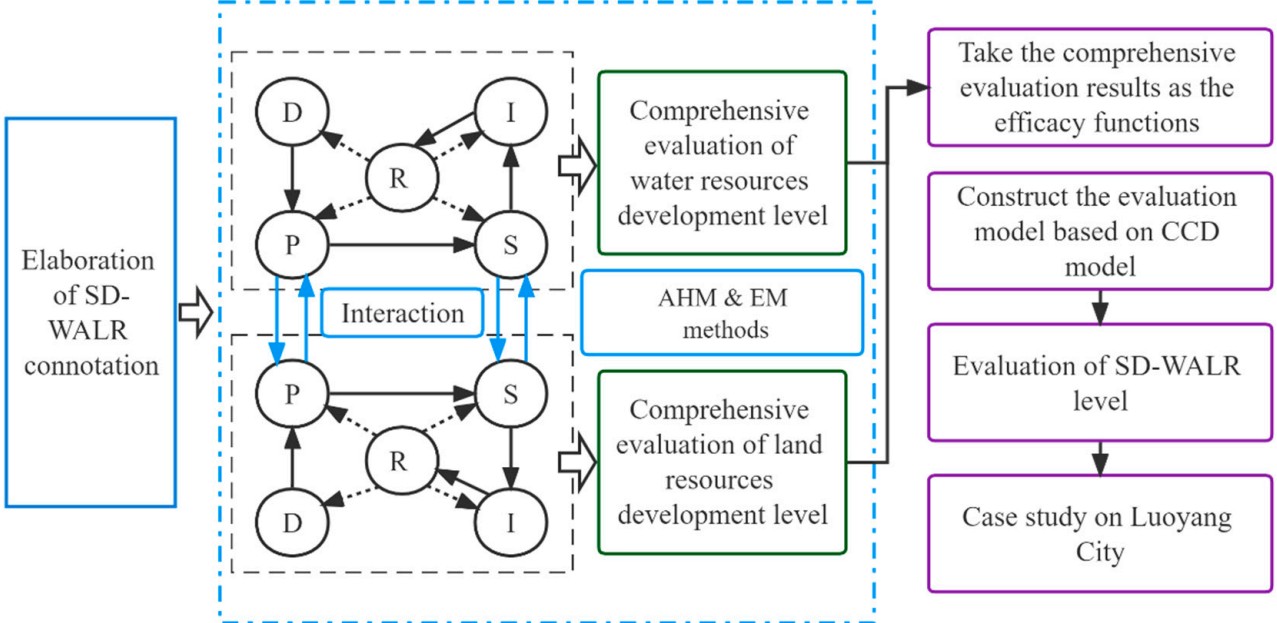

**Figure 1.** Research ideas and steps on the evaluation of the SD-WALRs.

## 2. Material and Methods

### 2.1. Study Area

Luoyang City (112°16′ E–112°37′ E, 34°32′ N–34°45′ N) is located in the west of Henan Province, China. It is a sub-center of the Central Plains Urban Agglomeration and has a high strategic position. The city covers 15,230 km$^2$ of land and has complex terrain. Mountainous and hilly regions account for a large proportion of its area, while plains only account for 13.8%. Land resources that are easy to utilize are still scarce. Meanwhile, water resources in Luoyang are insufficient. Since water resources in this city are mainly supplied by precipitation and groundwater, its distribution is remarkably spatiotemporally uneven and affected by the terrain and climate. Rapid population growth and social economic development have increased the pressure on the exploitation and utilization of WALRs. Moreover, the uneven distribution of water has also greatly restricted the utilization of land resources. The shortage of WALRs and difficult natural conditions for utilization limit the further development of Luoyang City. It is obvious that Luoyang must realize the SD-WALRs to overcome the shortage of resources and break the bottleneck of development. The geographical location of Luoyang and some spatial characteristics of its WALRs are shown in Figure 2. The original data used for this research were from the Henan Statistical Bureau (HSB 2010–2019), Luoyang Statistical Bureau (LSB 2010–2019), Luoyang Water Conservancy Bureau (LWCB 2010–2019) and "Luoyang City land use overall planning".

### 2.2. Effects of the SD-WALRs

Synergetic theory was first proposed by the German physicist Haken in 1969. It refers to the process in which various parts of a system or different systems cooperate to generate a new structure [20]. Moreover, the SD of systems emphasizes that the structure of subsystems would be more stable, the function would be more effective and the relationship of subsystems would be highly correlated through interaction. Therefore, the effects of the SD could be summarized into three parts based on the studies of Takayuki [21] and other scholars. First, the elements of subsystems fit each other and develop orderly. Second, the coupling relationship between subsystems is strong and the cooperation is efficient. Last but not least, the whole system adapts to the external eco-environment. It is notable that these three parts are only distinguished for convenience of understanding. In interpreting the effects of the SD, they are inseparable. The effects are shown schematically in Figure 3.

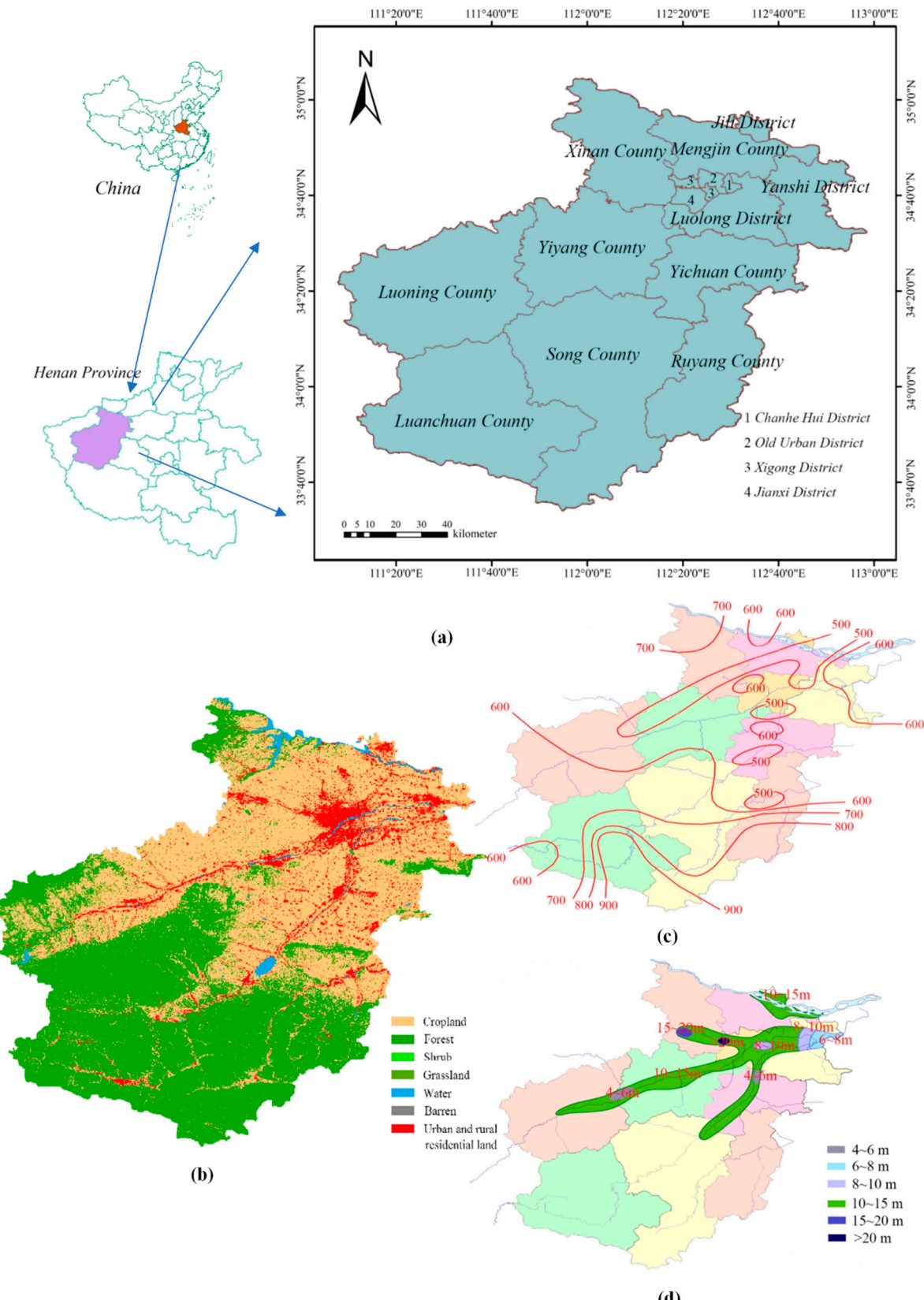

**Figure 2.** Geographical location of Luoyang (**a**), land type and distribution of Luoyang (**b**), isohyet of Luoyang in 2019 (**c**) and groundwater depth of Luoyang in 2019 (**d**).

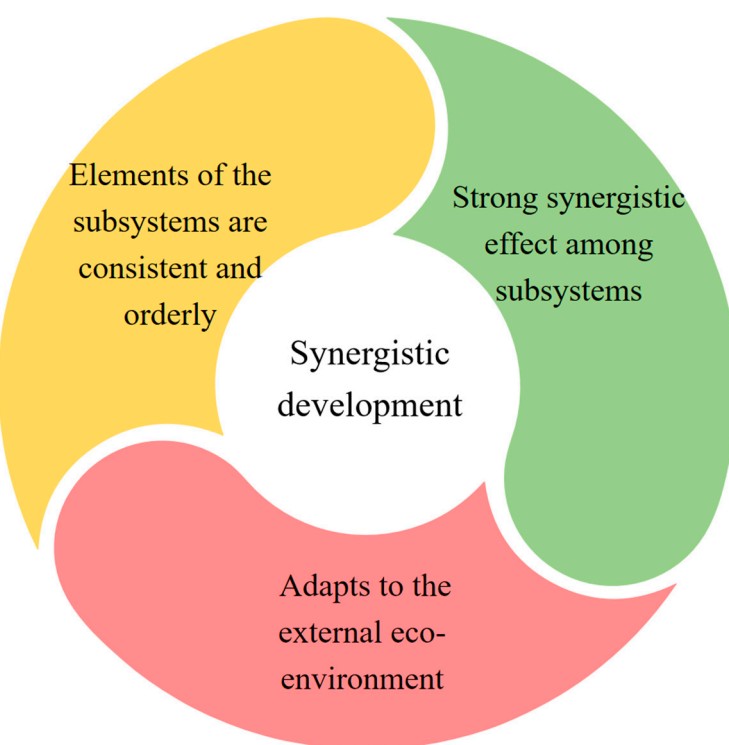

**Figure 3.** The effects of synergistic development.

Then, based on the above analysis and the relationship between resources and production activities, the effects of the SD-WALRs can be expressed as follows:

(1) The elements of the WALR subsystem match each other and develop in an orderly manner. The main goal is that the quantity and quality of WALRs can meet human needs, ensure production needs and maintain ecological stability. In addition, there are no adverse consequences, such as environmental deterioration, resource scarcity or a decline in people's living standards under the current utilization state. The structure and functions of the whole system are stable and developing.

(2) The second part says that the interaction between the two subsystems of WALRs is strong and coordinative. The essential feature is that the quantity and distribution of WALRs match. Furthermore, the SD-WALRs is mainly reflected in society and economy, which means that utilization and production of WALRs are in close combination, water use for various types of land can be guaranteed and the water use of various departments is efficient.

(3) The last part emphasizes the adaptability of WALRs to the eco-economic system's development and evolution. It refers to people being able to obtain higher eco-economic benefits in the current utilization state. Resources have great potential for exploration to support future development.

In summary, the evaluation method must meet several requirements: it can not only evaluate the subsystems' development levels and strengths of their mutual relationships accurately but also reflect the adaptability of WALRs to the external eco-environment.

### 2.3. Evaluation Index of the SD-WALR Subsystems Development State Based on the DPSIR Model

The DPSIR model is an improved version of the PSR (pressure–state–response) conceptual model proposed by the OECD (Organisation for Economic Co-operation and Development). It was first adopted by the European Environment Agency in 1993. Now, the model is often used to analyze environmental issues, sustainable development and other relevant problems because it can provide clear ideas for the structure or function analysis of complex systems and avoid missing decisive information [22,23]. In addition, it

can assist in selecting indexes of the evaluation model. Therefore, the DPSIR model can simplify the analysis process regarding complex interactions between various elements of the WALRs system, and scientifically select representative indexes that are pertinent and maneuverable [24]. The model is composed of five parts, namely, D (driving force), P (pressure), S (state), I (impact) and R (response). Considering the role of each part in the evaluation of the SD-WALRs on the basis of its effects given above, the relationship of each part is shown in Figure 4.

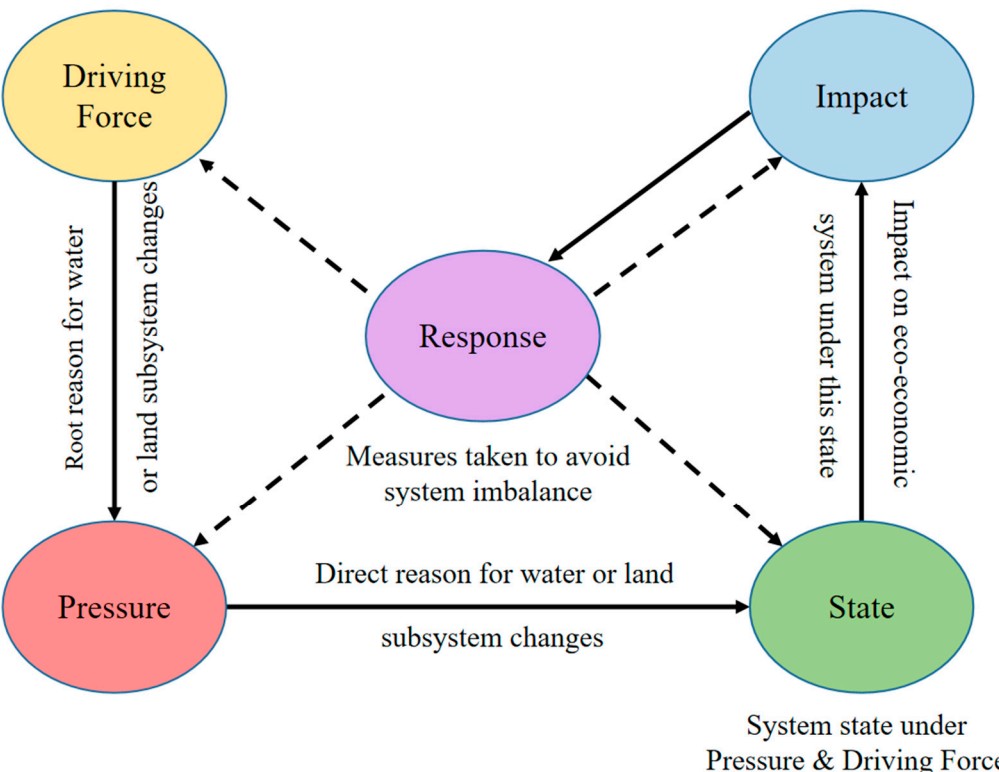

**Figure 4.** Relationship of each part of the DPSIR model in the SD-WALRs.

Based on the effects of the SD-WALRs, the index systems for evaluating the development level of WALR subsystems can be selected using the DPSIR model.

For the water or land resources subsystem, the main driving factors of system development are the natural and socio-economic driving forces. The occurrence of extreme natural conditions is random and not sequential [22], and synergistic development of WALRs is more concerned with the matching and coordination of water and land resources under the conditions of human activities. Therefore, in the short term, the socio-economic driving force should be the main driving factor used to evaluate the development level of WALRs. Population growth and social and economic development will affect the distribution and utilization pattern of regional WALRs. Therefore, the resident population, gross domestic product (GDP) and urbanization rate were selected to reflect the effect of the social and economy on the subsystems of WALRs. In addition, water pollution and land degradation indirectly affect the availability of regional resources. Water whose quality is worse than grade V will lose its usable value, where the higher the proportion of water worse than grade V, the lower the amount of water resources that will be available. Meanwhile, bare land cannot be directly utilized or requires a large amount of money to be invested in governance before developed. The higher the vegetation coverage, the smaller the degree of land degradation. Therefore, it is necessary to add the proportion of water worse than grade V and the ratio of vegetation coverage as evaluation indexes.

As a direct factor of subsystem changes, pressure comes from two aspects: one is the demand of social and economic development for the quantity and quality of resources, while the other is the competing demands for water and land resources in utilization. The

change in land-use structure will reallocate water resources. Meanwhile, water resources also constrain various land-based production activities. For the water resources subsystem, the water resources per capita, total water consumption in the region and volume of sewage were selected to reflect the pressure of its development. For the land resources, the land resources per capita, proportion of regional development and ecological index of cultivated land were chosen to reflect it. Since the competing demands from production activities for WALRs are mainly reflected in the structure of land and water use, corresponding indexes were added in the pressure part of the DPSIR model.

The state is the realistic reflection of the WALR subsystem under the driving force and pressure. The matching degree of resources is directly related to whether humans can realize sustainable utilization, which must be taken into account in the evaluation [25]. Moreover, effective development and utilization of resources can help humans to obtain greater eco-economic benefits, which are also the reflection of the function and structure stability. Therefore, water consumption per unit of industrial added value, irrigation water per unit area, and an urban or rural domestic comprehensive water quota can reflect the efficiency of industrial, agricultural and domestic water use. The proportion of effective irrigation area and per capita housing area can also reflect the land resource utilization status of various departments. The GDP per unit industrial area can also reflect the intensive use of the industrial land area.

The impact is the final result of the system's development and evolution. The orderly development of WALR subsystems causes the utilization of resources to be more efficient, which causes unit resources to produce greater value and conspicuously improves the living standards of human beings. Obviously, GDP per unit resource should be an important index to reflect the impact. Engel's coefficient is usually used to reflect the living standards of residents, which was also an indicator used in the model. Moreover, the water resources exploitation ratio and land diversity index can reflect the health degree of system development. They were also selected as evaluation indexes.

To achieve orderly development of the systems, humans must adjust their production intensity and mode, which is the response of human society regarding WALR subsystems [22]. People adapting land for cultivation or reclaiming land from the lake may cause soil erosion. Production activities will also generate sewage and solid wastes. In order to prevent the system's structure and functions from degrading, people have increased investment to control soil erosion and dispose of sewage and other wastes. Therefore, investment in a water conservancy project, the area of soil erosion controlled, unit land assets investment, and the proportion of sewage and solid wastes disposal were selected to measure the model's response part.

Based on the above analysis, the evaluation indexes of the development level of WALR subsystems were selected and shown in Tables 1 and 2.

### 2.4. Evaluation Model of the SD-WALRs

The level of the SD-WALRs was evaluated using the CCD model, which is based on the concept of the capacity coupling degree in physics [26,27]. If the synergistic development of the whole system is regarded as a multi-objective problem, the contribution of each element in a subsystem to the development is a sub-goal. The effect of each element on the synergistic development can be expressed by setting efficacy coefficients that are evaluated using the DPSIR model, and the efficacy functions can reflect the relationship of various efficacy coefficients. The CCD model sets efficacy functions to reflect the contribution of subsystems to the whole system development and evaluates the positive coupling effects of the subsystems' interaction using the coupling coordination degree. Then, the results can reflect whether the system has good coupling, as well as harmonious and orderly evolution [28]. The model is very suitable for analyzing the development trend of the WALR system and is very intuitive.

**Table 1.** Evaluation index system of the development level of water resources subsystems.

| Item | Index | Unit | Notes or Calculation Method | Item | Index | Unit | Notes or Calculation Method |
|---|---|---|---|---|---|---|---|
| Driving force | GDP (+) ($D_1$) | CNY $10^8$ | The drive for socio-economic evolution | State | Gini coefficient (−) ($S_2$) | / | Spatial matching degree of WALRs, calculation method from Li [9] |
| | Resident population (+) ($D_2$) | $10^4$ p | | | Irrigation water per unit area (+) ($S_3$) | $m^3$ | |
| | Urbanization rate (+) ($D_3$) | % | Urban/total population | | Water consumption per unit of industrial added value (−) ($S_4$) | $m^3$/CNY $10^4$ | Current water-use efficiency |
| | Proportion of water worse than grade V (−) ($D_4$) | % | Length of river worse than grade V/total river length | | Urban domestic comprehensive water quota (−) ($S_5$) | $m^3$/d | |
| Pressure | Cultivated area (−) ($P_1$) | $km^2$ | Pressure of land-use structure on water resources subsystem | | Rural domestic comprehensive water quota (−) ($S_6$) | $m^3$/d | |
| | Plantation area (−) ($P_2$) | $km^2$ | | Impact | GDP per unit water (+) ($I_1$) | CNY/$m^3$ | GDP/water consumption |
| Land-use structure | Industrial area (−) ($P_3$) | $km^2$ | | | Water resources exploitation ratio (−) ($I_2$) | % | Total water consumption/total regional water resources |
| | Residential building area (−) ($P_4$) | $km^2$ | | | Engel coefficient (−) ($I_3$) | % | Food expenditure/total personal expenditure |
| | Greenland aera (−) ($P_5$) | $km^2$ | | Response | Investment of water conservancy project (+) ($R_1$) | CNY $10^4$ | Total investment in water conservancy project construction and management |
| | Water resources per capita (+) ($P_6$) | $m^3$/p | Pressure of regional resources on water resources subsystem | | Ratio of sewage treatment (+) ($R_2$) | % | Measures taken to avoid system function degradation |
| | Total water consumption (−) ($P_7$) | $10^8$ $m^3$ | | | Area of soil erosion controlled (+) ($R_W$) | $km^2$ | |
| | Volume of sewage (−) ($P_8$) | $10^8$ $m^3$ | | | | | |
| State | Matching coefficient of WALRs (+) ($S_1$) | / | Quantity matching degree of WALRs, calculation method from Li [9] | | | | |

**Table 2.** Evaluation index system of synergistic development of land resources.

| Item | Index | Unit | Notes or Calculation Method | Item | Index | Unit | Notes or Calculation Method |
|---|---|---|---|---|---|---|---|
| Driving force | GDP (+) ($D_1$) | CNY $10^8$ | The drive for socio-economic evolution | State | Matching coefficient of WALRs (+) ($S_1$) | / | The same as Table 1 |
| | Resident population (+) ($D_2$) | $10^4$ p | | | Gini coefficient ($-$) ($S_2$) | / | |
| | Urbanization rate (+) ($D_3$) | % | | | Proportion of effective irrigation area (+) ($S_3$) | % | Present situation of land use |
| | Ratio of vegetation coverage (+) ($D_4$) | % | The drive for eco-environment evolution | | GDP per unit industrial area (+) ($S_4$) | CNY/$m^2$ | |
| Pressure | Water-use struct-ure | Proportion of agricultural water use ($-$) ($P_1$) | % | | Per capita housing area (+) ($S_5$) | $m^3$ | |
| | | Proportion of industrial water use ($-$) ($P_2$) | % | Pressure of water -use structure on land resources subsystem | | GDP per unit area (+) ($I_1$) | CNY/$m^3$ | GDP/land development area |
| | | Proportion of domestic water for residents ($-$) ($P_3$) | % | | Impact | Land diversity index (+) ($I_2$) | / | $H = -\sum_{i=1}^{n} p_i \ln p_i$ |
| | | Proportion of ecological water use ($-$) ($P_4$) | % | | | | | $H$: land diversity index; $p_i$: proportion of land type $i$; $n$: number of land-use types |
| | Land resources per capita (+) ($P_5$) | $m^2$/p | Pressure of regional resources on land resources subsystem | | Engel coefficient ($-$) ($I_3$) | % | The same as Table 1 |
| | Proportion of regional development ($-$) ($P_6$) | % | Construction land area/total area | Response | Unit land assets investment (+) ($R_1$) | CNY/$m^2$ | |
| | | | | | Area of soil erosion controlled (+) ($R_2$) | $km^2$ | Measures taken to avoid system function degradation |
| | Ecological index of cultivated land ($-$) ($P_7$) | kg/$m^2$ | (Total amount of pesticide, chemical fertilizer and agricultural film)/cultivated area | | Harmless treatment ratio of solid waste (+) ($R_3$) | % | |

Note: The raw data of all indicators are from Henan Statistical Bureau (HSB 2010–2019), Luoyang Statistical Bureau (LSB 2010–2019), Luoyang Water Conservancy Bureau (LWCB 2010–2019) and "Luoyang City land use overall planning".

According to the effects, the main determinants of the SD-WALRs under their mutual relationship are the development level of subsystems. This can be evaluated by setting comprehensive evaluation index systems. At the same time, close integration, good linkage and efficient interactions are also characteristics of the SD-WALRs. Therefore, the comprehensive evaluation index constructed above can be used as the efficacy functions of the CCD model. Then, the cooperation and interaction strength of the WALR subsystems in their respective development level can be represented using the coupling coordination degree. In this way, the level of the SD-WALRs can be evaluated. The model-constructing steps are mainly divided into the following two parts:

(1) A comprehensive evaluation of the development level of WALR subsystems.

Suppose $U_1$ and $U_2$ are the development levels of the two WALR subsystems, then:

$$U_{1(2)} = \sum z_{ij}\lambda_i \tag{1}$$

where $z_{ij}$ is the normalized value of index $i$ of the water or land resource subsystem in year $j$, which can be calculated via the min-max normalization (MMN) method using the original data, and $\lambda_i$ is the weight of index $i$.

Common weighting methods can be generally divided into two categories: one is subjective weighting methods represented by the expert grading (EG) method and the analytic hierarchy process (AHP) model, while the other is objective weighting methods, such as the entropy weight (EW) method and the criteria importance through intercriteria correlation (CRITIC) method. However, it is inevitable that the subjective weighting method could be affected by human cognition and experience, while the objective weighting method sometimes cannot take the importance of indexes in special situations into account because it is completely based on objective data. Therefore, this research made a combination and used the EW method and attribute hierarchical model (AHM) to weight the indexes. The AHM is an improved version of the AHP model, which has the advantages of the AHP model and omits the consistency check when using it [29]. Meanwhile, the EW method weights indicators based on the information entropy of the data [30]. Combining the AHM and EW methods to calculate weights can make the best use of the advantages and bypass the disadvantages of these two. The main steps are as follows:

Suppose $\lambda_i$ is the combined weight of the index $i$ of WALRs, which can be calculated using Formula (2):

$$\lambda_i = \frac{\omega_{AHM}\omega_{EW}}{\sum\limits_{i=1}^{n} \omega_{AHM}\omega_{EW}} \tag{2}$$

where $\omega_{AHM}$ and $\omega_{EW}$ are the weights found using the AHM and EW methods, respectively; $\omega_{AHM\text{-}i}$ can be obtained using Formulas (3) and (4):

$$\omega_{AHM-i} = \frac{2}{n(n-1)}\sum_{j=1}^{n} \mu_{ij} \tag{3}$$

$$\mu_{ij} = \begin{cases} \frac{ak}{ak+1} & (a_{ij} = k) \\ \frac{1}{ak+1} & (a_{ij} = \frac{1}{k}) \\ 0.5 & (a_{ij} = 1, i \neq j) \\ 0 & (a_{ij} = 1, i = j) \end{cases} \tag{4}$$

where $\omega_{AHM\text{-}i}$ is the weight of index $i$ calculated using the AHM; $\mu_{ij}$ is the relative attribute measure; $a_{ij}$ is the relative importance of factor $i$ to factor $j$, and its reciprocal is found according to Satty's original proposal; and the transformation relationship between $a_{ij}$ and $\mu_{ij}$ is shown in Formula (4). $k$ is the ratio of importance pairs determined using Satty's method [31]. $a$ is the attribute measure conversion parameter, which usually takes the value 1 or 2.

$\omega_{EW}$ can be obtained using Formulas (5)–(7):

$$\omega_{EW-i} = \frac{1 - H_j}{n - \sum\limits_{i=1}^{m} H_j} \tag{5}$$

$$H_j = -\frac{1}{\ln n} \sum_{i=1}^{n} y_{ij} \ln y_{ij} \tag{6}$$

$$y_{ij} = \frac{(1 + z_{ij})}{\sum\limits_{i=1}^{n} (1 + z_{ij})} \tag{7}$$

where $\omega_{EW-i}$ is the weight of index $i$ found using the EW method, $H_j$ is the entropy value of the indicator $i$ in year $j$ and $y_{ij}$ represents the proportion of the index $i$ in year $j$.

(2) Evaluation model of the SD-WALRs

Taking $U_1$ and $U_2$ as the efficacy function of the CCD model, the evaluation model of the SD-WALRs is as follows:

$$C = 2 \times \frac{\sqrt{U_1 \times U_2}}{U_1 + U_2} \tag{8}$$

$$T = \alpha U_1 + \beta U_2 \tag{9}$$

$$SD = \sqrt{C \times T} \tag{10}$$

where $C$ is the coupling degree of the WALR subsystems and ranges from 0 to 1. When $C = 0$, there is no interaction between the two subsystems, and the extent of coupling between subsystems increases with $C$. $T$ is the coordination degree of the WALRs, which is used to show the positive coupling extent of the subsystems. $\alpha$ and $\beta$ are undetermined coefficients of the subsystems' contributions; because water and land resources are equally important for human development, $\alpha$ and $\beta$ were both set equal to 0.5. $SD$ is the index of the SD-WALR level, which represents the degree of coupling and the harmonious and orderly evolution of the WALRs.

With the study of system coupling or synergy by [28,32,33], combined with the effects of the SD-WALRs in this paper, the level of the SD-WALRs can be classified according to Table 3.

**Table 3.** Classification of the synergistic development level.

| Classes | Range of Index | Types of SD-WALRs |
|---|---|---|
| Unbalanced recession | $0 < SD \leq 0.1$ | Extremely unbalanced recession |
| | $0.1 < SD \leq 0.2$ | Severely unbalanced recession |
| | $0.2 < SD \leq 0.3$ | Intermediate unbalanced recession |
| | $0.3 < SD \leq 0.4$ | Mildly unbalanced recession |
| Transition stage | $0.4 < SD \leq 0.5$ | On the verge of recession |
| | $0.5 < SD \leq 0.6$ | Reluctantly synergetic development |
| | $0.6 < SD \leq 0.7$ | Primary synergetic development |
| Synergetic development | $0.7 < SD \leq 0.8$ | Intermediate synergetic development |
| | $0.8 < SD \leq 0.9$ | Good synergetic development |
| | $0.9 < SD \leq 1$ | High-quality synergetic development |

## 3. Results

Since the required data has not been fully released since 2020, the latest 10 years that are available (2010~2019) were selected as the research period, which could not only avoid the loss of timeliness of the research but also reflect the changing trend of the development level. The original data of the water and land resources subsystem indexes

used in the research were obtained from the website (http://www.henan.gov.cn/ and http://shuili.ly.gov.cn/?m=home&c=Lists&a=index&tid=87) (accessed on 10 May 2022) The weights of indicators found using three methods based on Formulas (2)–(7) are shown in Table 4.

**Table 4.** Weights of indicators found using three methods.

| | Index Weight of the Water Resources Subsystem | | | | Index Weights of the Land Resources Subsystem | | |
|---|---|---|---|---|---|---|---|
| Items | $\omega_{AHM}$ | $\omega_{EW}$ | $\lambda_i$ | Items | $\omega_{AHM}$ | $\omega_{EW}$ | $\lambda_i$ |
| $D_1$ | 0.0278 | 0.0358 | 0.0236 | $D_1$ | 0.0278 | 0.0352 | 0.0210 |
| $D_2$ | 0.0369 | 0.0506 | 0.0444 | $D_2$ | 0.0369 | 0.0497 | 0.0393 |
| $D_3$ | 0.0153 | 0.0386 | 0.0140 | $D_3$ | 0.0153 | 0.0379 | 0.0124 |
| $D_4$ | 0.0200 | 0.0255 | 0.0121 | $D_4$ | 0.0200 | 0.0490 | 0.0210 |
| $P_1$ | 0.0323 | 0.0450 | 0.0345 | $P_1$ | 0.0212 | 0.0348 | 0.0158 |
| $P_2$ | 0.0167 | 0.0449 | 0.0178 | $P_2$ | 0.0459 | 0.0591 | 0.0581 |
| $P_3$ | 0.0448 | 0.0396 | 0.0422 | $P_3$ | 0.0181 | 0.0413 | 0.0160 |
| $P_4$ | 0.0192 | 0.0301 | 0.0137 | $P_4$ | 0.0117 | 0.0587 | 0.0147 |
| $P_5$ | 0.0188 | 0.0361 | 0.0161 | $P_5$ | 0.0881 | 0.0448 | 0.0845 |
| $P_6$ | 0.0542 | 0.0418 | 0.0538 | $P_6$ | 0.0770 | 0.0624 | 0.1029 |
| $P_7$ | 0.0658 | 0.0414 | 0.0647 | $P_7$ | 0.0380 | 0.0364 | 0.0296 |
| $P_8$ | 0.0483 | 0.0567 | 0.0651 | $S_1$ | 0.0461 | 0.0469 | 0.0463 |
| $S_1$ | 0.0442 | 0.0477 | 0.0501 | $S_2$ | 0.0461 | 0.0348 | 0.0344 |
| $S_2$ | 0.0442 | 0.0354 | 0.0372 | $S_3$ | 0.0905 | 0.0680 | 0.1318 |
| $S_3$ | 0.0731 | 0.0482 | 0.0837 | $S_4$ | 0.0838 | 0.0309 | 0.0555 |
| $S_4$ | 0.0869 | 0.0281 | 0.0580 | $S_5$ | 0.0335 | 0.0365 | 0.0262 |
| $S_5$ | 0.0258 | 0.0330 | 0.0202 | $I_1$ | 0.0755 | 0.0342 | 0.0553 |
| $S_6$ | 0.0258 | 0.0589 | 0.0361 | $I_2$ | 0.0375 | 0.0380 | 0.0305 |
| $I_1$ | 0.0755 | 0.0364 | 0.0653 | $I_3$ | 0.0375 | 0.0385 | 0.0309 |
| $I_2$ | 0.0375 | 0.0317 | 0.0282 | $R_1$ | 0.0495 | 0.0460 | 0.0488 |
| $I_3$ | 0.0375 | 0.0392 | 0.0349 | $R_2$ | 0.0500 | 0.0497 | 0.0532 |
| $R_1$ | 0.0495 | 0.0574 | 0.0675 | $R_3$ | 0.0500 | 0.0669 | 0.0717 |
| $R_2$ | 0.0500 | 0.0476 | 0.0565 | | | | |
| $R_3$ | 0.0500 | 0.0505 | 0.0600 | | | | |

The combined weights had the advantages of subjective and objective weighting methods. They started from the entropy value and modified the weights of some indexes according to the special situation in the study area, such as the structure of the tertiary, industry and policy management. Then, the annual data of the study area were normalized using the MMN method. Based on Formula (1), the results of the development level of the WALRs in Luoyang from 2010 to 2019 are shown in Table 5, and the value change trends of the various DPSIR parts are shown in Figures 5 and 6.

**Table 5.** Evaluation results of the development level of the WALR subsystems. Unit: $10^{-2}$.

| Item | Year | 2010 | 2011 | 2012 | 2013 | 2014 | 2015 | 2016 | 2017 | 2018 | 2019 |
|---|---|---|---|---|---|---|---|---|---|---|---|
| Water resources subsystem | D | 0.00 | 1.59 | 2.31 | 3.15 | 3.90 | 5.14 | 6.33 | 6.71 | 8.51 | 9.42 |
| | P | 19.70 | 17.50 | 11.42 | 7.28 | 13.01 | 17.36 | 16.48 | 13.53 | 11.81 | 9.41 |
| | S | 10.46 | 15.51 | 15.81 | 17.7 | 17.53 | 12.94 | 8.25 | 9.66 | 12.43 | 15.61 |
| | I | 4.07 | 5.03 | 4.08 | 1.83 | 4.04 | 6.30 | 7.76 | 9.42 | 10.53 | 10.79 |
| | R | 2.16 | 6.50 | 6.14 | 10.95 | 1.11 | 4.81 | 12.32 | 11.77 | 11.58 | 12.00 |
| | U | 36.39 | 46.12 | 39.77 | 40.92 | 39.60 | 46.55 | 51.13 | 51.08 | 54.85 | 57.22 |
| Land resources subsystem | D | 0.02 | 0.59 | 1.24 | 2.08 | 2.99 | 4.10 | 5.21 | 6.51 | 8.13 | 9.37 |
| | P | 23.68 | 19.65 | 19.74 | 19.92 | 22.05 | 16.01 | 10.08 | 11.92 | 9.68 | 8.84 |
| | S | 4.91 | 10.97 | 9.34 | 4.46 | 8.41 | 19.33 | 17.64 | 20.91 | 25.36 | 26.03 |
| | I | 1.11 | 2.51 | 3.45 | 3.17 | 4.62 | 6.54 | 7.80 | 9.50 | 10.67 | 11.43 |
| | R | 6.56 | 0.40 | 1.41 | 6.33 | 1.57 | 6.21 | 8.19 | 10.71 | 11.28 | 12.04 |
| | U | 36.28 | 34.14 | 35.18 | 35.95 | 39.64 | 52.19 | 48.91 | 59.54 | 65.12 | 67.7 |

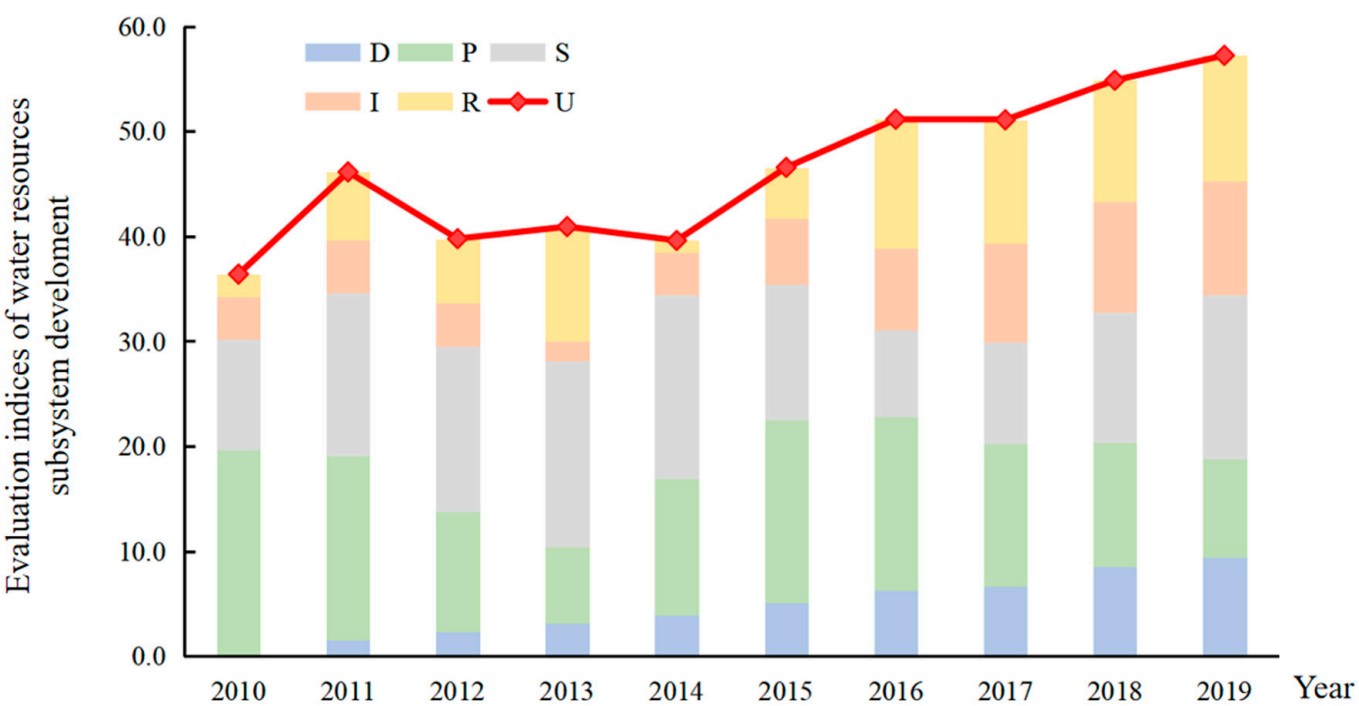

**Figure 5.** Trends of the development level of the water resources.

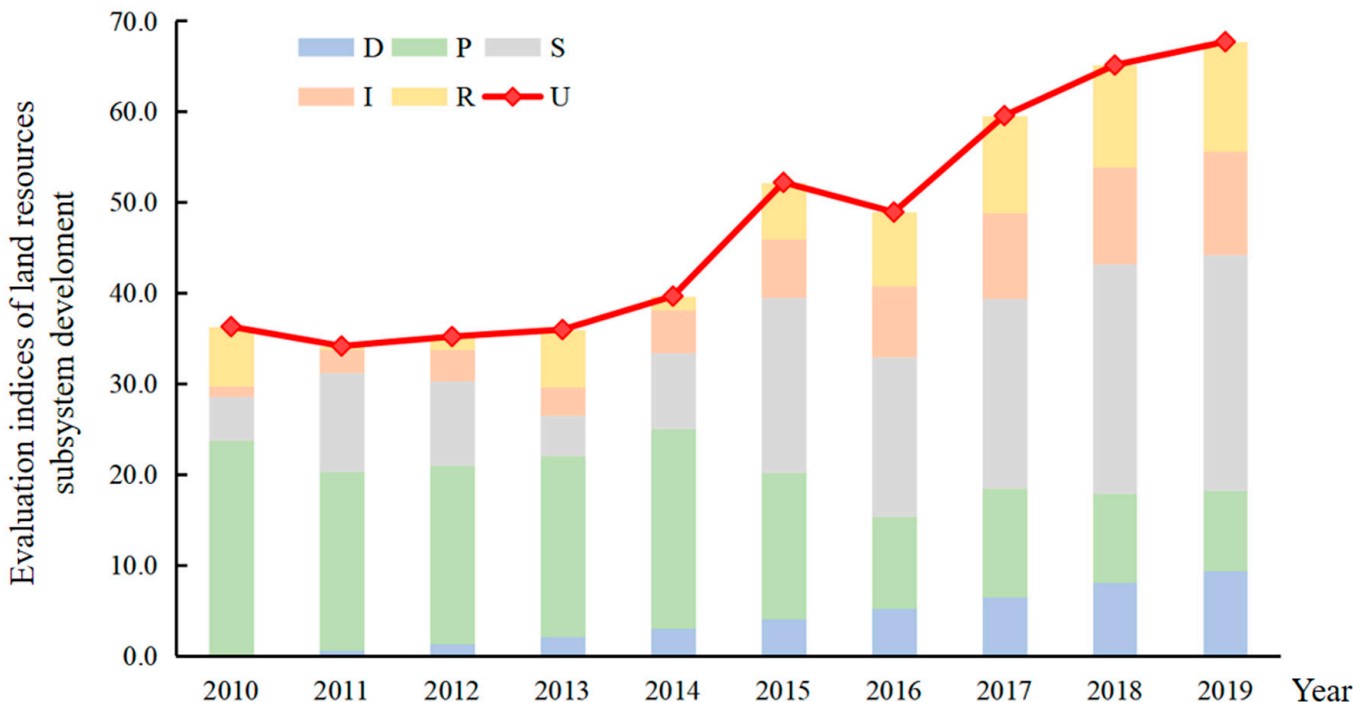

**Figure 6.** Trends of the development level of the land resources.

C, T and SD were calculated by Formulas (8)–(10) based on the results of Table 4. According to the classification in Table 3, the evaluation results from 2010 to 2019 are displayed in Table 6 and the trends of the SD-WALRs are given in Figure 7.

**Table 6.** Evaluation results of the SD of water and land resources.

| Year | C | T | SD | Level of the SD-WALRs |
|---|---|---|---|---|
| 2010 | 0.999 | 0.363 | 0.603 | Primary synergetic development |
| 2011 | 0.989 | 0.401 | 0.630 | Primary synergetic development |
| 2012 | 0.998 | 0.375 | 0.612 | Primary synergetic development |
| 2013 | 0.998 | 0.384 | 0.619 | Primary synergetic development |
| 2014 | 0.999 | 0.396 | 0.629 | Primary synergetic development |
| 2015 | 0.998 | 0.494 | 0.702 | Intermediate synergetic development |
| 2016 | 0.999 | 0.500 | 0.707 | Intermediate synergetic development |
| 2017 | 0.997 | 0.553 | 0.743 | Intermediate synergetic development |
| 2018 | 0.996 | 0.600 | 0.773 | Intermediate synergetic development |
| 2019 | 0.996 | 0.625 | 0.789 | Intermediate synergetic development |

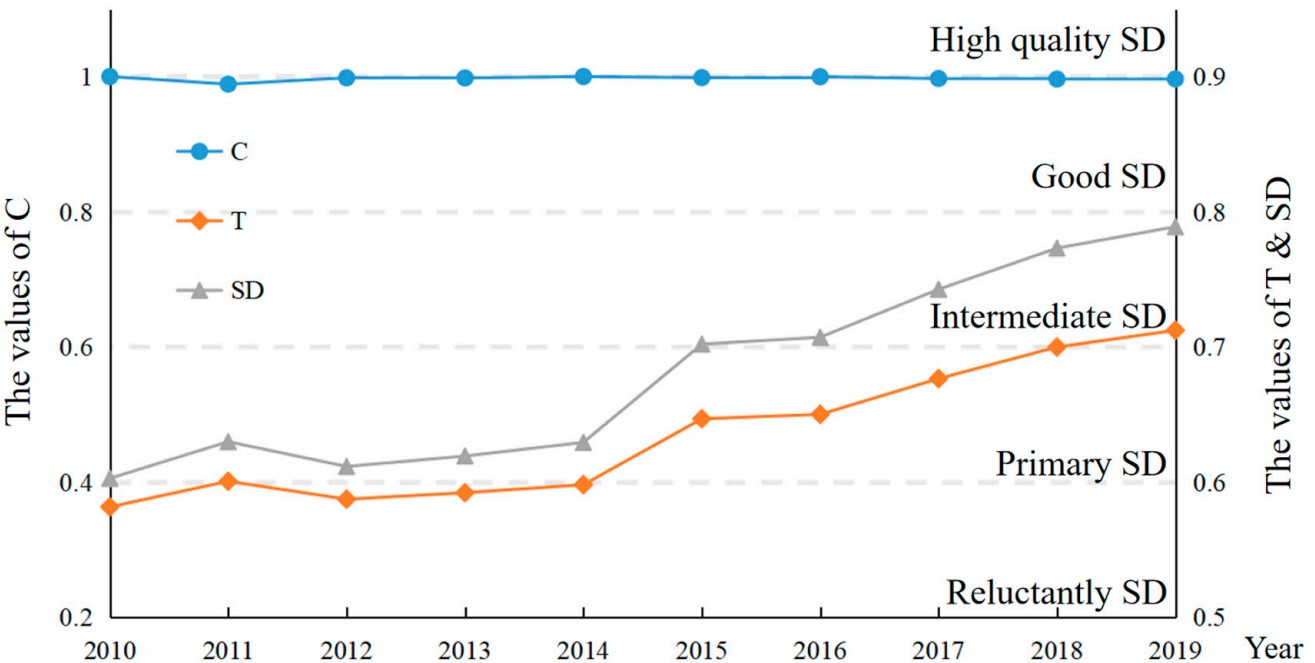

**Figure 7.** Trends of the C, T and SD and the evaluation classifications of the SD.

## 4. Discussion

### 4.1. Evaluation Results of the WALR Subsystems Development

It can be seen from Table 5 and Figures 5 and 6 that the value of the WALR subsystem development level showed a fluctuating upward trend during the study period, which indicated that the internal elements of the WALR subsystem gradually developed from "disorder" to "order". The values of pressure and state changed greatly. Compared with 2010, the pressure values of the water subsystems decreased from 0.197 to 0.094, and the pressure values of land subsystems decreased from 0.237 to 0.088, respectively, while the value of state increased from 0.041 and 0.011 to 0.179 and 0.114, respectively. The values of impact and response also greatly improved compared with the beginning of the study period. In order to analyze the reasons for their changes, the radar figures of normalized data of the indexes in 2011, 2014 and 2019 are shown in Figure 8, and the trends of some indexes with remarkable changes from 2010 to 2019 are shown in Figure 9.

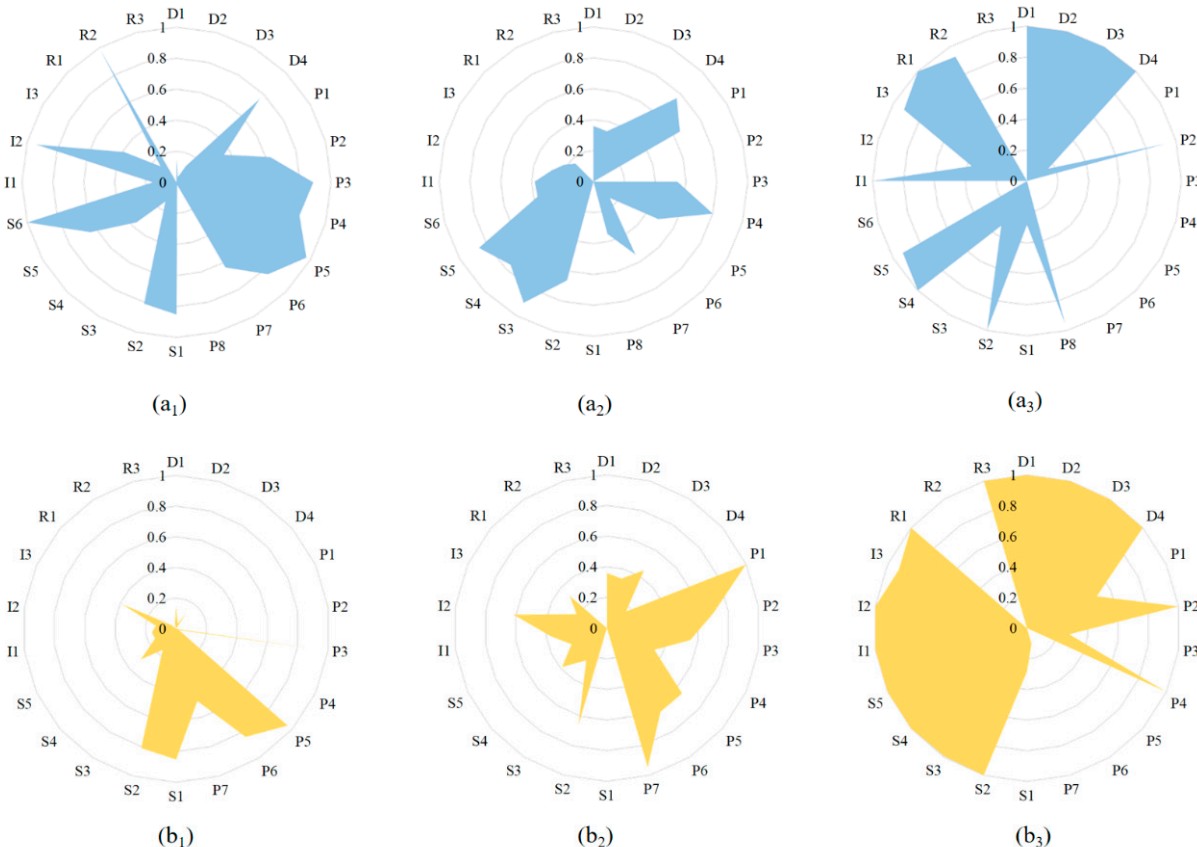

**Figure 8.** Normalized data of the WALR indexes: (**a₁**,**a₂**,**a₃**) normalized data of the water resources subsystem indexes in 2010, 2014 and 2019, respectively; (**b₁**,**b₂**,**b₃**) normalized data of the land resources subsystem indexes in the corresponding years.

Combined with the original data and Figures 8 and 9, it can be seen that the increase in the population and urbanization process enhanced the driving force of the system. However, the pressure of the system decreased and the state value increased significantly. This was mainly helped by the effective implementation of various measures for WALR protection and efficient utilization in Luoyang in recent years, such as the control of soil erosion and vigorously promoting the popularization of water-saving production technology and water-saving appliances. The government continuously increased investment in the development, utilization and treatment of WALRs. From 2010 to 2019, the accumulated soil erosion in Luoyang reached 2100 km$^2$ and the ratio of solid waste and sewage treatment was maintained above 95%. The changes in the state indexes, such as the increase in the proportion of effective irrigation area, the decrease in the average irrigation water consumption per area and water consumption industrial added value per CNY 10 thousand, reflected the fact that the utilization efficiency of the WALRs in Luoyang gradually improved, and the development of the WALRs was more in line with the current situation of social and economic development. Figure 9c shows that the area of the industrial land and green land significantly increased compared with the beginning of the study period, where many unused and uncultivated areas were converted to industrial land and green land. The total regional water consumption changed little, the proportion of industrial water consumption decreased moderately, and the proportion of ecological water consumption increased remarkably, which not only reflected the improvement in industrial water-use efficiency but also reflected the fact that the water-use structure of Luoyang was more reasonable.

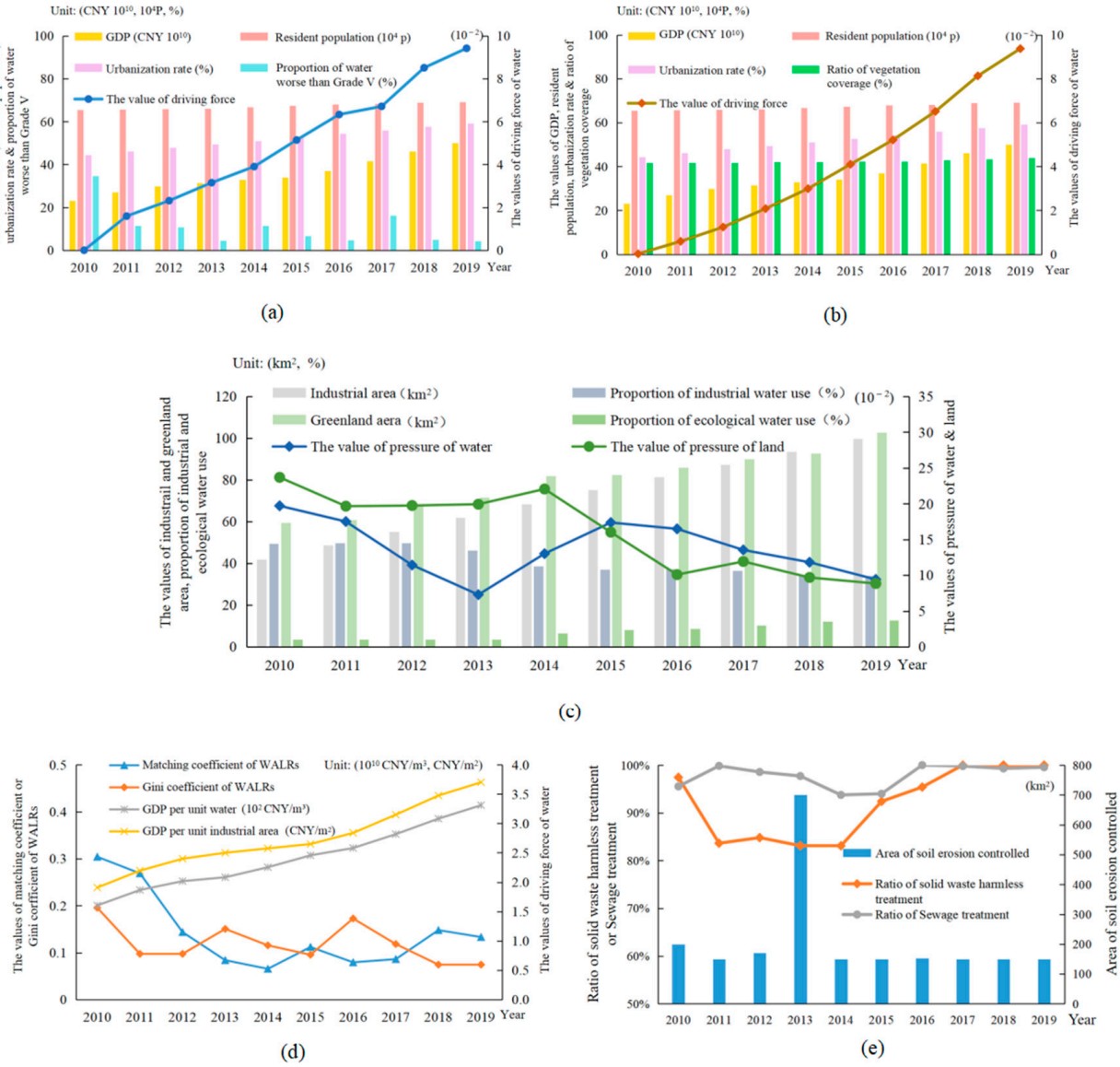

**Figure 9.** Part of the indexes with remarkable changes from 2010 to 2019: (**a**,**b**) main driving force indexes of water and land respectively, (**c**) trends of two representative resources utilization states, (**d**) trends of the matching degree and GDP per unit resources, and (**e**) measures taken to avoid degradation by humans.

It is worth noting that the response values of the water resources subsystem fluctuated greatly from 2014 to 2015, and for the land resources subsystem, this phenomenon appeared in 2011, 2012 and 2014. From Figure 9e, the fluctuation was mainly related to the reduction in the soil-erosion-controlled area and the ratio of sewage and waste treatment in these years. Compared with 2013, the soil-erosion-controlled area in 2014 decreased by 550 km$^2$, and the sewage and solid waste treatment ratio was the lowest in the study period. In addition, the improvement of the state value was highly related to the development and utilization of the WALRs, and the matching degree of the total amount and distribution of the WALRs was stable for ten years. In this study, the method used for calculating the matching degree of the WALRs was the same as in Li's research [9]. As shown in Figure 9d, the Gini coefficient reflecting the spatial matching degree of the WALRs in Luoyang was stable at about 0.1 and the spatial matching degree of the WALRs was classified as good. However, from 2010 to 2014, the system was not in a high-level coordinated state. The spatial and temporal matching of water resources is an important factor that restricts

regional sustainable development. However, we cannot judge whether the WALRs reached a high level of SD only from the degree of spatial matching.

### 4.2. Evaluation Results of the SD-WALRs

Table 6 and Figure 7 show that the coupling degree of the WALRs in Luoyang was stable and close to 1 during the study period. The value of the comprehensive coordination index was low in the early years but it rose greatly later, showing a fluctuating upward trend as a whole. This indicated that there was a strong interaction between water and land resources in Luoyang, and the cooperative relationship between the water and land resources subsystems was constantly strengthened. From 2010 to 2019, the index of the SD-WALRs in Luoyang increased from 0.603 to 0.789, which meant an increase from the primary synergetic development level to intermediate synergetic development. The overall trend showed that the subsystem developed from "high coupling but low synergetic" to "high coupling and synergetic". Some of the indexes in the impact part, such as the GDP per water unit, GDP per unit land area, land diversity index, development and utilization ratio of WALRs, and Engel's coefficient improved greatly. This reflected the fact that the WALRs had a positive impact on the outside of the system at this synergetic level.

Lv [18] evaluated the interaction between WALRs in Guiyang City. Because his research did not elucidate the effects of the SD-WALRs, the interaction between the WALR subsystems was evaluated using the coupling coordination degree. According to his method, the indexes of evaluation were selected from the utilization and protection states of the WALR subsystems; then, the level of interaction could be evaluated using the CCD model. By using his method, the trends from his research and this study are shown in Figure 10. It can be clearly seen that the values of the coupling degree and coupling coordination degree evaluated using Lv's method were smaller than from this research, the coordinated degree values were close and the trends of these three indexes were almost consistent. The reason for the difference is that impact of the interaction of the WALRs on the respective development of subsystems was not considered in the index-selection process. In the analysis of this research, the development level of water or land resources subsystems under their mutual relationship is vital for the SD-WALRs. For a more scientific and accurate evaluation, indexes that can reflect the matching degree of WALRs, the structure of development and utilization, and the impact on socio-economy can be selected for use in the evaluation systems.

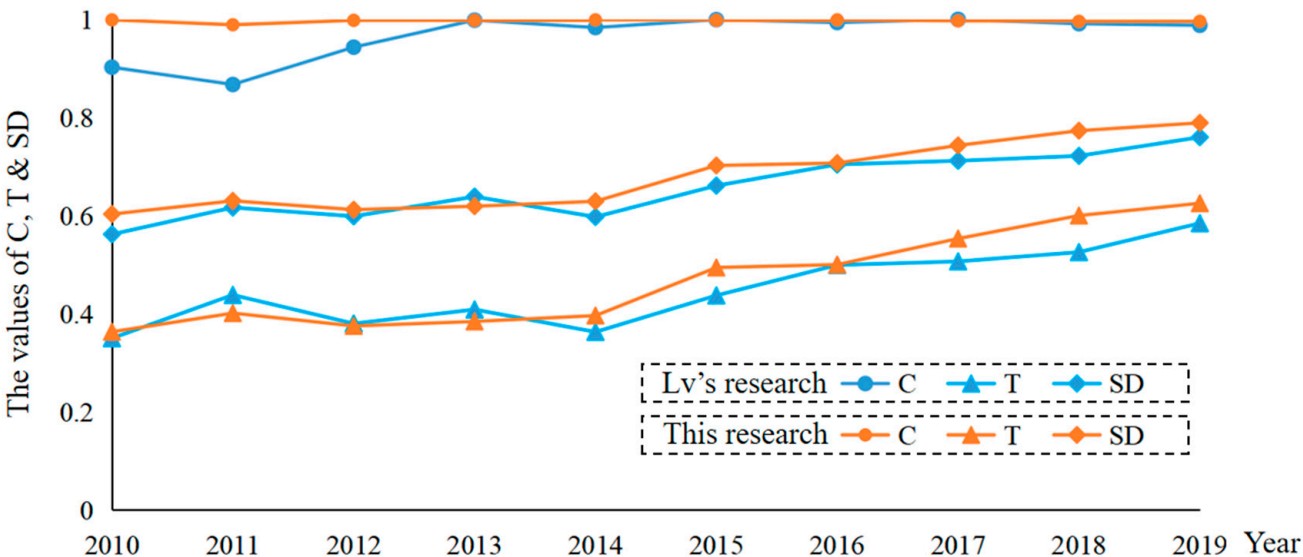

**Figure 10.** Evaluation results of this research and Lv's method.

## 5. Conclusions

Based on the synergetic theory, this study comprehensively analyzed the effects of the SD-WALRs. The evaluation index systems reflecting the development level of the WALR subsystem were selected using the DPSIR model. Then, the weights of indexes were assigned through the combination of subjective and objective weighting methods. Finally, the evaluation model of the SD-WALRs was constructed with the help of the CCD model. Taking Luoyang City as an example, the case verification was completed. The evaluation results showed that Luoyang City maintained a strong interaction between the WALRs for many years. In recent years, Luoyang City scientifically formulated land-use planning and continuously strengthened water resource management. With the increase in the area of soil erosion controlled, the promotion of efficient water-saving technologies and the improvement of waste treatment, the level of subsystems development and the cooperation effect were improved. This showed that the development and utilization of WALRs will be more scientific and reasonable. Meanwhile, it was found that the growth in the development level of the land resource subsystem was higher than the water resource subsystem, where water resources will restrict the further development and utilization of land resources. To achieve a higher level of synergetic development of the WALRs, Luoyang must further strengthen the optimization and management of water resources subsystems by optimizing the water-use structure, improving water-use efficiency and increasing investment in water conservancy project construction. However, there are still some problems. The DPSIR model is a simplification of the interaction relationship of various elements in the system, which can describe the overall coordinated evolution trend of the system in a macro way. However, there is insufficient explanation of the interaction relationship, evolution mode and feedback mechanism between various elements in the system. In addition, the evaluation model in this study was only an annual evaluation of the level and change trend. The detailed process of system evolution was not depicted, which still needs further research.

**Author Contributions:** All authors contributed to this study's conception and design. Conceptualization: C.L.; methodology: C.L. and W.X.; formal analysis and investigation: M.L.; writing—original draft preparation: C.L. and W.X.; writing—review and editing: C.L., W.X., S.W. and Y.H.; supervision: S.W. All authors have read and agreed to the published version of the manuscript.

**Funding:** This research was funded by the Key Technologies Research and Development Program of China (2021YFC3000204) and the National Natural Science Foundation of China (NSCF-52079125).

**Data Availability Statement:** Data of Luoyang City used in research were from the statistical yearbook of Henan Province, the water resources bulletin of Henan Province and Luoyang, etc. The specific data source can be accessed at http://www.henan.gov.cn/ (accessed on 10 May 2022) and http://shuili.ly.gov.cn/?m=home&c=Lists&a=index&tid=87 (accessed on 10 May 2022).

**Acknowledgments:** This research was funded by the Key Technologies Research and Development Program of China (2021YFC3000204) and the National Natural Science Foundation of China (NSCF-52079125). The authors are grateful to colleagues and friends who shared their meteorological and hydrological data with us. We also thank the reviewers for insightful comments that improved an earlier version of this manuscript.

**Conflicts of Interest:** The authors declare that they have no competing interests or other interests that might be perceived to influence the results and/or discussion reported in this paper.

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
