# Peer review of "Evaluation of Synergetic Development of Water and Land Resources Based on a Coupling Coordination Degree Model"

_water, doi:10.3390/w15081491_

Round 1

Reviewer 1 Report

In this paper, the authors evaluate the pressures and tradeoffs between water and land resources using a DPSIR framework with Luoyang City as a case study. The paper has the potential to shed insight into how various socioeconomic and natural forces impact the availability and use of water and land-based resources and could provide valuable information for resource planners and policymakers. Unfortunately, I don’t think the paper goes far enough to adequately address those questions current form. While the paper is well organized I often had difficulty understanding the logic of certain methodological decisions and suggest adding significant detail about the datasets being used and justification for the metric derived from them. The results from the DPSIR analysis also have the potential to shed interesting insights into how socioeconomic and natural factors have impacted resource sustainability over the study period but I don’t think those were adequately explored. In the discussion section, the authors begin to explore how each component of the DPSIR framework has changed as a result of the underlying metrics, just stating that one of the components has changed without exploring the reason or implications for that change leaves the reader asking “so what?”. Overall I would suggest major revisions before this article can be considered for publication. I would also suggest proofreading as there are quite a few grammatical errors throughout the manuscript.

Minor comments.

Line 1: I’d recommend introducing the WALR acronym in the main text before using

Lines 36-28: I found this statement a little confusing. What do you mean by “water resources are the driving forces of the geologic cycle”?

Lines 43: I found this statement a little confusing too: “Besides, matching degree and 44 reciprocity are also remarkable influence factors...”. factors of what?

Line 83: It’s not clear to me what you mean by “connotation” and I’m wondering if a different word would be more appropriate since this seems a like central component of your design. Would “coevolution” be appropriate?

Line 131: slid -> slide?

Line 132-133: Incomplete sentence, please revise

Line 135: coordinative -> coordinated?

Line 149: Specify OECD as the Organisation for Economic Co-operation and Development

Lines 166-167: I’m not sure I understand what you’re saying here, but to me, it seems that natural factors also have the potential to be strong, rapid, and highly variable in space. For example, extreme events such as droughts, heat waves, and floods could all have dramatic impacts on WALR

Lines 73-175: Your metrics aren’t very clear at this point so it’s unclear what the implications are for an increase in “Grade V” or  the “ratio of vegetation coverage” or if those changes are justified. I’d suggest including a section describing your evaluation indices.

Lines 176-180: This text should most likely be in the introduction.

Line 186: Again, please explicitly state what indices you’re using

Lines 212-213: Tables 1 and 2 helped clarify the indices you’re using in your DPSIR analysis, but more detail is needed for each. For example, what are the spatial and temporal scales of each index? How were they acquired? No source is given. Were there any transformations done?

Line 218: What is the CCD model? Please add some detail for the reader.

Line 251: Should “EM” be “EW”?

Line 283: Could you describe the implications of the “D” metric?

Lines 289-293: This should probably be stated in the methods section where you describe the indices.

Figures 5 & 6: Can you add labels to the x and y axes?

Figure 9: Please add axes labels here as well, and what are the units for the “value” lines? Also, it looks like GDP is increasing in panel a, but static in panel b. Is this correct? A result of the scale?

Line 346-348: As someone not familiar with the region I find it interesting that both industrial and green land areas increased significantly over this time. Out of curiosity, what land use types are being converted to industrial and green land use?

Line 362: Please provide a citation for “Li Hui’s research”.

Author Response

We are grateful for your insightful review. The provided suggestions have contributed substantially to improving the paper. Accordingly, we have made efforts to significantly revise the manuscript. While, the main point of our article is how to evaluate the level of synergetic development of WALR by CCD model based on synergetic theory. Due to the word limit of the article, We can only show normalized values of all indicators for three important periods in Figure 8 and analyze some factors with significant changes. Detailed modifications please see the attachment.

Reviewer 2 Report

Review of “Evaluation of Synergetic Development of Water and Land Resources Based on Coupling Coordination Degree Model”

The authors evaluated the synergetic development of interaction between water and land resources in Luoyang with DPSIR model. The connotation of synergetic development has also been elucidated. However, significant results and conclusions have not been provided. The significance of the study also need to be explained. I would suggest acceptance subject to major revisions. My questions that might help authors improve the paper are:

(1)   How to assess the level of synergetic with the evaluation method of DPSIR?

(2) The significance of evaluating SD level of WALR should be explained.

(3) What is the weakness or issues in practice in the studied area?

(4) What is the mean of coupling development and synergetic development? What is the difference between “high coupling but low synergetic development” and “high synergetic development”?

(5) Errors exist in this version of the manuscript, like “On the other hand, Water is required…” in line 38.

(6) Line 42, the sentence “It is obvious that quantity and quality of WALR can directly affect eco-economic benefits for human” is not clear. How to understand the quantity and quality of interaction.

(7) Line 80, what is the significance of connotation analysis of SD-WALR?

(8) CCD model is used for the evaluation of the study framework, what is the advantages of this technique?

(9) In “Conclusions” part, more conclusion points of the studied case should be provided, and recommendations for Luoyang should be given. The conclusion part is of serious disqualification.

Author Response

We are grateful for your insightful review. The provided suggestions have contributed substantially to improving the paper. Accordingly, we have made efforts to significantly revise the manuscript. Detailed modification please see the attachment.

Round 2

Reviewer 2 Report

The manucsirpt can be accepted after the revisions.